# Stochastic Expectation Maximization with Variance Reduction

**Jianfei Chen**[†]**, Jun Zhu**[†,*]**, Yee Whye Teh**[‡] **and Tong Zhang**[§]

[†] Dept. of Comp. Sci. & Tech., BNRist Center, State Key Lab for Intell. Tech. & Sys.,
Institute for AI, THBI Lab, Tsinghua University, Beijing, 100084, China
[‡] Department of Statistics, University of Oxford
[§] Tencent AI Lab
{chenjian14@mails,dcszj@}.tsinghua.edu.cn
y.w.teh@stats.ox.ac.uk; tongzhang@tongzhang-ml.org

## Abstract

Expectation-Maximization (EM) is a popular tool for learning latent variable models, but the vanilla batch EM does not scale to large data sets because the whole data set is needed at every E-step. Stochastic Expectation Maximization (sEM) reduces the cost of E-step by stochastic approximation. However, sEM has a slower asymptotic convergence rate than batch EM, and requires a decreasing sequence of step sizes, which is difficult to tune. In this paper, we propose a variance reduced stochastic EM (sEM-vr) algorithm inspired by variance reduced stochastic gradient descent algorithms. We show that sEM-vr has the same exponential asymptotic convergence rate as batch EM. Moreover, sEM-vr only requires a constant step size to achieve this rate, which alleviates the burden of parameter tuning. We compare sEM-vr with batch EM, sEM and other algorithms on Gaussian mixture models and probabilistic latent semantic analysis, and sEM-vr converges significantly faster than these baselines.

## 1 Introduction

Latent variable models are an important class of models due to their wide applicability across machine learning and statistics. Examples include factor analysis in psychology and the understanding of human cognition [32], hidden Markov models for modelling sequences, e.g. speech and language [29], and DNA [15], document and topic models [17, 4] and mixture models for density estimation and clustering [26]. Expectation Maximization (EM) [12] is a basic tool for maximum likelihood estimation for the parameters in latent variable models. It is an iterative algorithm with two steps: an E-step which calculates the expectation of sufficient statistics under the latent variable posteriors given the current parameters, and an M-step which updates the parameters given the expectations.

With the phenomenal growth in big data sets in recent years, the basic batch EM (bEM) algorithm in [12] is quickly becoming infeasible because the whole data set is needed at every E-step. Cappé and Moulines [6] proposed a stochastic EM (sEM) algorithm for exponential family models, which reduces the time complexity for the E-step by approximating the full-batch expectation with an exponential moving average over minibatches of data. sEM has been adopted in many applications including natural language processing [24], topic modeling [16, 14] and hidden Markov models [5]. However, sEM has a slow asymptotic convergence rate due to the high variance of each update. Unlike the original batch EM (bEM), which converges exponentially fast near a local optimum, the distance towards a local optimum only decreases at the rate $O(1/\sqrt{T})$ for sEM, where $T$ is the

---

[*]corresponding author

number of iterations. Moreover, sEM requires a decreasing sequence of step sizes to converge. The decay rate of step sizes is often difficult to tune.

Recently, there has been much progress in accelerating stochastic gradient descent (SGD) by reducing the variance of the stochastic gradients, including SAG, SAGA and SVRG [22, 20, 11]. These algorithms achieve better convergence rates by utilizing infrequently computed batch gradients as control variates. Such ideas have also been brought into gradient-based Bayesian learning algorithms, including stochastic variational inference [25], as well as stochastic gradient Markov-chain Monte-Carlo [13, 8, 7] (SGMCMC).

In this paper, we develop a variance reduced stochastic EM algorithm (sEM-vr). In each epoch, that is, a full pass through the data set, our algorithm computes the full batch expectation as a control variate, and uses this to reduce the variance of minibatch updates in that epoch. Let $E$ be the number of epochs and $M$ be the number of minibatch iterations per epoch. We show that near a local optimum, our algorithm, with a constant step size, enjoys a convergence rate of $O((M^{-1} \log M)^{E/2})$ to the optimum. Like bEM, our convergence rate is exponential with respect to the number of epochs, and is asymptotically faster than sEM. We also show that our algorithm converges globally with a constant step size, under stronger assumptions. Note that leveraging variance reduction ideas in sEM is not straightforward, since sEM is not a stochastic gradient descent algorithm but rather a stochastic approximation [21] algorithm. In particular, the proof techniques we utilize are different than those in stochastic gradient descent algorithms. We demonstrate our algorithm on Gaussian mixture models and probabilistic latent semantic analysis [18]. sEM-vr achieves significantly faster convergence comparing with sEM, bEM, and other gradient-based and Bayesian algorithms.

## 2 Background

We review batch and stochastic EM algorithms in this section. Throughout the paper we focus on exponential family models with tractable E- and M-steps, which stochastic EM [6] is designed for.

### 2.1 EM Algorithm

The EM algorithm is designed for models with some observed variable $x$ and hidden variable $h$. We assume an exponential family joint distribution $p(x, h; \theta) = b(x, h) \exp\{\eta(\theta)^\top \phi(x, h) - A(\theta)\}$ parameterized by $\theta$. Given a data set of $N$ ($\gg 1$) observations $X = \{x_i\}_{i=1}^N$, we want to obtain a maximum likelihood estimation (MLE) of the parameter $\theta$, by maximizing the log marginal likelihood $\mathcal{L}(\theta) := \sum_{i=1}^N \log p(x_i; \theta) = \sum_{i=1}^N \log \int_{h_i} p(x_i, h_i; \theta) d\theta$, where the variables $(x_i, h_i)$ are i.i.d. given $\theta$. Denote $H = \{h_i\}_{i=1}^N$. Batch expectation-maximization (bEM) [12] optimizes the log marginal likelihood $\mathcal{L}(\theta)$ by constructing a lower bound of it:

$$\mathcal{L}(\theta) \geq Q(\theta; \hat{\theta}) - E_{p(H|X;\hat{\theta})}\left[\log p(H|X, \hat{\theta})\right], \tag{1}$$

$$Q(\theta; \hat{\theta}) := \mathbb{E}_{p(H|X;\hat{\theta})}[\log p(X, H; \theta)] = N\left(\eta(\theta)^\top F(\hat{\theta}) - A(\theta)\right) + \text{constant}, \tag{2}$$

where we define $F(\hat{\theta}) := \frac{1}{N} \sum_{i=1}^N f_i(\hat{\theta})$ as the full-batch expected sufficient statistics, and where $f_i(\hat{\theta}) := \mathbb{E}_{p(h_i|x_i;\hat{\theta})}[\phi(x_i, h_i)]$ is the expected sufficient statistics conditioned on observed datum $x_i$.

Let $\hat{\theta}_e$ be the estimated parameter at iteration or epoch $e$, where each epoch is a complete pass through the data set. In the E-step, bEM tightens the bound in Eq. (1) by setting $\hat{\theta} = \hat{\theta}_e$, and computes the expected sufficient statistics $F(\hat{\theta}_e)$. In the M-step, bEM finds a maximizer $\hat{\theta}_{e+1}$ of the lower bound with respect to $\theta$, by solving the optimization problem $\text{argmax}_\theta\{\eta(\theta)^\top F(\hat{\theta}) - A(\theta)\}$. The solution is denoted as $R(F(\hat{\theta}))$, and is assumed to be tractable. In summary, the bEM updates can be written simply as

$$\text{E-step: compute } F(\hat{\theta}_e), \quad \text{M-step: let } \hat{\theta}_{e+1} = R(F(\hat{\theta}_e)). \tag{3}$$

The algorithm is also applicable to maximum *a posteriori* (MAP) estimation of parameters, with a conjugate prior $p(\theta; \alpha) = \exp\{\eta(\theta)^\top \alpha - A(\theta)\}$ with the hyperparameter $\alpha$. Instead of $\mathcal{L}(\theta)$, MAP maximizes $\mathcal{L}(\theta) + \log p(\theta; \alpha) \geq N\eta(\theta)^\top \left(\alpha/N + F(\hat{\theta})\right) - NA(\theta) + \text{constant}$, and we still apply Eq. (3), but with $f_i(\hat{\theta}) := \alpha/N + \mathbb{E}_{p(h_i|x_i;\hat{\theta})}[\phi(x_i, h_i)]$ instead.

## 2.2 Stochastic EM Algorithm

When the data set is large, that is, $N$ is large, computing $F(\hat{\theta}_t)$ in the E-step is too expensive because it needs a full pass though the entire data set. Stochastic EM (sEM) [6] avoids this by maintaining an exponentially moving average $\hat{s}_t$ as an approximation of the full average $F(\hat{\theta}_t)$. At iteration $t$, sEM picks a single random datum $i$, and updates:

$$\text{E step: } \hat{s}_{t+1} = (1 - \rho_t)\hat{s}_t + \rho_t f_i(\hat{\theta}_t), \quad \text{M step: } \hat{\theta}_{t+1} = R(\hat{s}_{t+1}),$$

where $(\rho_t)$ is a sequence of step sizes that satisfy $\sum_t \rho_t = \infty$ and $\sum_t \rho_t^2 < \infty$. We deliberately choose different iteration indices $e$ and $t$ for bEM and sEM to emphasize their different time complexity per iteration. In practice, sEM can take a minibatch of data instead of a single datum per iteration, but we stick to a single datum for cleaner presentation. The two sEM updates can be rolled into a single update

$$\hat{s}_{t+1} = (1 - \rho_t)\hat{s}_t + \rho_t f_i(\hat{s}_t). \tag{4}$$

where for simplicity we have overloaded the notation with $f_i(s) := f_i(R(s))$. This first maps $s$, which can be interpreted as the estimated mean parameter of the model, into the parameters $\theta = R(s)$, before computing the required expected sufficient statistics $f_i(\theta)$ under the posterior given observation $x_i$. Which of the two definitions should be clear from the type of its argument and we feel this helps reduce notational burden on the reader. We similarly overload $F(s) := F(R(s))$ and $\mathcal{L}(s) := \mathcal{L}(R(s))$ accordingly, so we can also write bEM updates (Eq. 3) as simply $\hat{s}_{e+1} = F(\hat{s}_e)$. Intuitively, we want to find a stationary point $s_*$ under bEM iterations, i.e., $s_* = F(s_*)$. We can view bEM as a fixed-point algorithm, and sEM as a Robbins-Monro [30] algorithm to solve the equation $s_* = F(s_*)$.

Because of the cheap updates, sEM can converge faster than bEM on large data sets in the beginning. However, due to the variance of the estimator $\hat{s}_t$, sEM has a slower asymptotic convergence rate than bEM for finite data sets. Specifically, let $s_* = F(s_*)$ be a stationary point, Cappe and Monlines [6] showed that $\mathbb{E} \|\hat{s}_T - s_*\|^2 = O(\rho_T)$ for sEM, which is at best $O(T^{-1})$ since $\sum_t \rho_t = \infty$. In contrast, Dempster et al. [12] showed that bEM converges as $\|\hat{s}_E - s_*\|^2 \leq (1 - \lambda)^{-2E} \|\hat{s}_0 - s_*\|$, where $1 - \lambda \in [0, 1)$ is a constant that is defined in Sec. 3.3. As long as the data set is finite, the exponential rate of bEM is faster than sEM. [2] Moreover, sEM needs a decreasing sequence of step sizes to converge, whose decay rate is difficult to tune.

## 3 Variance Reduced Stochastic Expectation Maximization

In this section, we describe a variance reduced stochastic EM algorithm (sEM-vr), and develop the theory for its convergence. sEM-vr enjoys an exponential convergence rate with a constant step size.

## 3.1 Algorithm Description

We run the algorithm for $E$ epochs and $M$ minibatch iterations per epoch, so that there are $T := ME$ iterations in total. For simplicity we choose $M = N$ and use minibatches of size 1, though our analysis is not limited to this case. Each epoch has the same time complexity as bEM. We index iteration $t$ in epoch $e$ as $e, t$. Let $\hat{s}_{e,t}$ be the estimated sufficient statistics at iteration $e, t$. Starting from the initial estimate $\hat{s}_{0,0}$, sEM-vr performs the following updates in epoch $e$,

---

**Stochastic EM with Variance Reduction**
1. Compute $F(\hat{s}_{e,0})$, and save $F(\hat{s}_{e,0})$ as well as $\hat{s}_{e,0}$
2. For each iteration $t = 1, \ldots, M$, randomly sample a datum $i$, and update

$$\hat{s}_{e,t+1} = (1 - \rho)\hat{s}_{e,t} + \rho \left[ f_i(\hat{s}_{e,t}) - f_i(\hat{s}_{e,0}) + F(\hat{s}_{e,0}) \right]. \tag{5}$$

3. Let $\hat{s}_{e+1,0} = \hat{s}_{e,M}$.

---

Let $\mathbb{E}_{e,t}$ and $\text{Var}_{e,t}$ be the expectation and variance over the random index $i$ in iteration $e, t$. Comparing Eq. (5) with Eq. (4), we observe that the sEM and sEM-vr updates have the same expectation

$\mathbb{E}_t [\hat{s}_{t+1}] = (1 - \rho)\hat{s}_t + \rho F(\hat{s}_t)$. However their variances are different: sEM has $\mathrm{Var}_t [\hat{s}_{t+1}] = \rho_t^2 \mathrm{Var}_t[f_i(\hat{s}_t)]$, while sEM-vr has $\mathrm{Var}_{e,t} [\hat{s}_{e,t+1}] = \rho^2 \mathrm{Var}_{e,t} [f_i(\hat{s}_{e,t}) - f_i(\hat{s}_{e,0})]$. If the algorithm converges, i.e., the sequence $(\hat{s}_{e,t})$ converges to a point $s_*$, and $f_i(\cdot)$ is continuous, the variance of sEM-vr will converge to zero, while that of sEM will remain positive. Therefore, sEM-vr has asymptotically smaller variance than sEM, and we will see that this leads to better asymptotic convergence rates.

The time complexity of sEM-vr per epoch is the same as bEM and sEM, with a constant factor up to 3, for computing $f_i(\hat{s}_{e,t}), f_i(\hat{s}_{e,0})$ and $F(\hat{s}_{e,0})$. The space complexity also has a constant factor up to 3, for storing $\hat{s}_{e,0}$ and $F(\hat{s}_{e,0})$ along with $\hat{s}_{e,t}$. In practice, the difference is less than 3 times because the time and space costs for other aspects of the methods are the same, e.g. data storage.

## 3.2 Related Works

A possible alternative to sEM is Titterington's online algorithm [33], which replaces the exact M-step with a gradient ascent step to optimize $Q(\theta; \hat{\theta})$, where the gradient is multiplied with the inverse Fisher information of $p(x, h; \theta)$. Titterington's algorithm is locally equivalent to sEM [6]. However, as argued by Cappé and Moulines [6], Titterington's algorithm has several issues, including the Fisher information being expensive to compute in high dimensions, the need for explicit matrix inversion, and that the updated parameters are not guaranteed to be valid. Moreover, leveraging variance reduced stochastic gradient algorithms [20, 22, 11] for Titterington's algorithm is not straightforward as the Fisher information matrix changes with $\theta$. Zhu et al. has proposed a variance reduced stochastic gradient EM algorithm [39]. There are also some theoretical analysis of EM algorithm for high dimensional data [3, 35].

Instead of performing point estimation of parameters, Bayesian inference algorithms, including variational inference (VI) and Markov-chain Monte-Carlo (MCMC), can also be adopted, to infer the posterior distribution of parameters. Variance reducing techniques have also been applied to these settings, including smoothed stochastic variational inference (SSVI) [25] and variance reduced stochastic gradient MCMC (VRSGMCMC) algorithms [13, 8, 7]. However, convergence guarantees for SSVI have not been developed, while VRSGMCMC algorithms are typically much slower than sEM-vr due to the intrinsic randomness of MCMC. For example, the time complexity to converge to an $\epsilon$-precision in terms of the 2-Wasserstein distance of the true posterior and the MCMC distribution is $O(N + \kappa^{3/2}\sqrt{d}/\epsilon)$, where $\kappa$ is a condition number and $d$ is the dimensionality of the parameters [7].

## 3.3 Local Convergence Rate

We analyze the local convergence rate of a sequence $\{\hat{s}_{e,t}\}$ of sEM-vr iterates to a stationary point $s_*$ with $s_* = F(s_*)$. Let $\theta_* := R(s_*)$ be the natural parameter corresponding to the mean parameter $s_*$.

**Theorem 1.** *If*

> *(a) The Hessian $\nabla^2 \mathcal{L}(\theta_*)$ is negative definite, i.e., $\theta_*$ is a strict local maximum of $\mathcal{L}(\theta_*)$.*
> *(b) $\forall i$, $f_i(s)$ is $L_f$-Lipschitz continuous, and $F(s)$ is $\beta_f$-smooth.*
> *(c) $\forall e, t, \|\hat{s}_{e,t} - s_*\| < \lambda/\beta_f$, where $1 - \lambda$ is the maximum eigenvalue of $J_* := \partial F(s_*)/\partial s_*$.*

*Then, for any step size $\rho \leq \lambda/(32L_f^2)$, we have*

$$\mathbb{E} \|\hat{s}_{E,0} - s_*\|^2 \leq \left[\exp\left(-M\lambda\rho/4\right) + 32L_f^2\rho/\lambda\right]^E \|\hat{s}_{0,0} - s_*\|^2. \tag{6}$$

*In particular, if $\rho = \rho_* := 4\log(M/\kappa^2)/(\lambda M)$, where $\kappa^2 := 128L_f^2/\lambda^2$, then we have*

$$\mathbb{E} \|\hat{s}_{E,0} - s_*\|^2 \leq \left[\left(1 + \log(M/\kappa^2)\right)\kappa^2/M\right]^E \|\hat{s}_{0,0} - s_*\|^2. \tag{7}$$

**Remarks.** Assumption (a) follows directly from the original EM paper (Theorem 4) [12]. [12] analyzed the convergence only in an infinitesimal neighbourhood of $s_*$, while Assumption (c) gives an explicit radius of convergence. Assumption (b) is new and required to control the variance and radius of convergence. Note also that we analyse the convergence of the mean parameters, while [12] analysed that for parameters. However they are equivalent if $R(s)$ is Lipschitz continuous. In Appendix A.1 we show that negative definite $\nabla^2 \mathcal{L}(\theta_*)$ in Assumption (a) implies that $\lambda > 0$ in Assumption (c).

*Proof.* We first analyze the convergence behavior at a specific epoch $e$, and omit the epoch index $_e$ for concise notations. We further denote $\Delta_t := \hat{s}_t - s_*$ for any $t$. By Eq. (5),

$$\mathbb{E}_t \|\Delta_{t+1}\|^2 = \mathbb{E}_t \|(1-\rho)\hat{s}_t + \rho F(\hat{s}_t) - s_* + \rho\left[f_i(\hat{s}_t) - f_i(\hat{s}_0) - F(\hat{s}_t) + F(\hat{s}_0)\right]\|^2$$
$$= \|(1-\rho)\hat{s}_t + \rho F(\hat{s}_t) - s_*\|^2 + \rho^2 \mathbb{E}_t \|f_i(\hat{s}_t) - f_i(\hat{s}_0) - F(\hat{s}_t) + F(\hat{s}_0)\|^2, \qquad (8)$$

where the second equality is due to $\mathbb{E}_t\left[f_i(\hat{s}_{e,t}) - f_i(\hat{s}_{e,0}) + F(\hat{s}_{e,0})\right] = F(\hat{s}_{e,t})$. We have

$$\|(1-\rho)\hat{s}_t + \rho F(\hat{s}_t) - s_*\|^2 = \|(1-\rho)\Delta_t + \rho(F(\hat{s}_t) - s_*) + \rho J_* \Delta_t - \rho J_* \Delta_t\|^2$$
$$\leq \left[\|(1-\rho)\Delta_t + \rho J_* \Delta_t\| + \rho \|F(\hat{s}_t) - s_* - J_* \Delta_t\|\right]^2$$
$$\leq \left[(1-\rho\lambda)\|\Delta_t\| + (\rho/2)\beta_f\|\Delta_t\|^2\right]^2 = \left[1 - \rho\left(\lambda - \beta_f\|\Delta_t\|/2\right)\right]^2 \|\Delta_t\|^2$$
$$\leq (1-\rho\lambda/2)^2 \|\Delta_t\|^2 \leq (1-\rho\lambda/2)\|\Delta_t\|^2, \qquad (9)$$

where the second line utilizes triangular inequality, the third line utilizes $\|(1-\rho)I + \rho J_*\| \leq 1 - \rho + \rho(1-\lambda) = 1 - \rho\lambda$, where $\|\cdot\|$ is the $\ell_2$ operator norm, and the smoothness in (b), which implies $\|F(\hat{s}_t) - s_* - J_*(\hat{s}_t - s_*)\| \leq (\beta_f/2)\|\hat{s}_t - s_*\|^2$. The last line utilizes (c).

By (b), $F$ is $L_f$-Lipschitz and $\forall i, f_i - F$ is $2L_f$-Lipschitz continuous. Therefore

$$\mathbb{E}_t \|f_i(\hat{s}_t) - f_i(\hat{s}_0) - F(\hat{s}_t) + F(\hat{s}_0)\|^2 \leq 4L_f^2 \|\hat{s}_t - \hat{s}_0\|^2 \leq 8L_f^2(\|\Delta_t\|^2 + \|\Delta_0\|^2). \qquad (10)$$

Combining Eq. (8, 9, 10), and utilizing our assumption $\rho \leq \lambda/(32L_f^2)$, we have

$$\mathbb{E}\|\Delta_{t+1}\|^2 \leq \left(1 - \rho\lambda/2 + 8\rho^2 L_f^2\right)\|\Delta_t\|^2 + 8\rho^2 L_f^2\|\Delta_0\|^2 \leq (1-\rho\lambda/4)\|\Delta_t\|^2 + 8\rho^2 L_f^2\|\Delta_0\|^2.$$

We get Eq. (6, 7) by analyzing the sequence $a_{t+1} \leq (1-\epsilon\rho)a_t + c\rho^2 a_0$, where $a_t = \mathbb{E}\|\Delta_t\|^2$, $\epsilon = \lambda/4$ and $c = 8L_f^2$. The analysis is in Appendix B. $\qquad \square$

**Comparison with bEM:** As mentioned in Sec. 2.2, bEM has $\mathbb{E}\|\hat{s}_E - s_*\|^2 \leq (1-\lambda)^{2E}\|\hat{s}_0 - s_*\|^2$. The distance decreases exponentially for both bEM and sEM-vr, but at different speeds. If $M$ is large, sEM-vr (Eq. 7) converges much faster than bEM because $\left(1 + \log(M/\kappa^2)\right)\kappa^2/M \ll (1-\lambda)^2$, thanks to its cheap stochastic updates.

**Comparison with sEM:** As mentioned in Sec. 2.2, sEM has $\mathbb{E}\|\hat{s}_T - s_*\|^2 = O(T^{-1})$, which is not exponential, and is asymptotically slower than sEM-vr. The key difference is we can bound the variance term for sEM-vr by $\|\hat{s}_t - \hat{s}_0\|^2$ in Eq. (10), so the variance goes to zero as $(\hat{s}_{e,t})$ converges. The advantage of sEM-vr over sEM is especially significant when $E$ is large. Moreover, sEM requires a decreasing sequence of step sizes to converge [6], which is more difficult to tune comparing with the constant step size of sEM-vr.

### 3.4 Global Convergence

Theorem 1 only considers the case near a local maximum of the log marginal likelihood. We now show that under stronger assumptions, there exists a constant step size, such that sEM-vr can globally converge to a stationary point $s_* = F(s_*)$, one with $\nabla \mathcal{L}(s_*) = 0$ [12].

**Theorem 2.** *Suppose*

> *(a) The natural parameter function $\eta(\theta)$ is $L_\eta$-Lipschitz, and $f_i(s)$ is $L_f$-Lipschitz for all $i$,*
> *(b) for any $x$ and $h$, $\log p(x, h; \theta)$ is $\gamma$-strongly-concave w.r.t. $\theta$.*

*Then for any constant step size $\rho < \gamma/\left(M(M-1)L_\eta L_f\right)$, sEM-vr converges to a stationary point, starting from any valid sufficient statistics vector $\hat{s}_{0,0}$.*

A sufficient condition for (b) is the exponential family is canonical, i.e., $\eta(\theta) = \theta$, and we want the MAP estimation instead of MLE, where the prior $\log p(\theta)$ is $\gamma$-strongly-concave. We leave the proof in Appendix C. The idea is first show that sEM-vr is a generalized EM (GEM) algorithm [36], which improves $\mathbb{E}[Q(\theta; \hat{\theta})]$ after each epoch, and then apply Wu's convergence theorem for GEM [36].

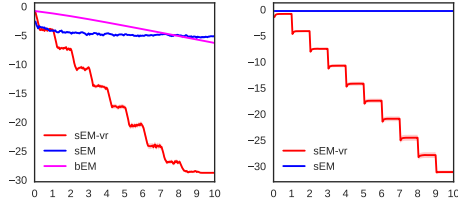

Figure 1: Toy Gaussian Mixture. Left: $\log_{10} \mathbb{E} \left\| \hat{\mu}_t - \mu_* \right\|^2$, Right: $\log_{10} \mathrm{Var}_t[\hat{\mu}_t]/\rho_t^2$, X-axis: number of epochs.

Table 1: Statistics of datasets for pLSA. k=thousands, m=millions.

| Data set | $D$ | $V$ | $|\mathcal{I}|$ |
|---|---|---|---|
| NIPS [1] | 1.5k | 12k | 1.93m |
| NYTimes [1] | 0.3m | 102k | 99m |
| Wiki [38] | 3.6m | 8k | 524m |
| PubMed [1] | 8.1m | 141k | 731m |

# 4 Applications and Experiments

We demonstrate the application of sEM-vr on a toy Gaussian mixture model and probabilistic latent semantic analysis.

## 4.1 Toy Gaussian Mixture

We fit a mixture of two Gaussians, $p(x|\mu) = 0.2\mathcal{N}(\mu, 1) + 0.8\mathcal{N}(-\mu, 1)$, with a single unknown parameter $\mu$. Let $X = \{x_i\}_{i=1}^N$ be the data set, and $h_i \in \{1, 2\}$ be the cluster assignment of $x_i$. We write $h_{ik} := \mathbb{I}(h_i = k)$ as a shortcut, where $\mathbb{I}(\cdot)$ is the indicator function. The joint likelihood is $p(X, H|\mu) \propto \exp\{\sum_i \sum_k h_{ik} \log \mathcal{N}(x_i; \mu_k, 1)\} \propto \exp\{\sum_i \eta(\mu)^\top \phi(x_i, h_i)\}$, where the natural parameter $\eta(\mu) = (\mu, -\mu, -\mu^2/2, \mu^2/2)$ and the sufficient statistics $\phi(x_i, h_i) = (x_i h_{i1}, x_i h_{i2}, h_{i1}, h_{i2})$. Let $\gamma_{ik}(\mu) = p(h_i = k|x_i, \mu) \propto \pi_i \mathcal{N}(x_i; \mu_k, 1)$ for $k \in \{1, 2\}$ be the posterior probabilities. The expected sufficient statistics $f_i(\mu) = \mathbb{E}_{p(h_i, x_i|\mu)} \phi(x_i, h_i) = (x_i \gamma_{i1}(\mu), x_i \gamma_{i2}(\mu), \gamma_{i1}(\mu), \gamma_{i2}(\mu))$, and $F(\mu) = 1/N \sum_i f_i(\mu)$. The mapping from sufficient statistics to parameters is $R(s) = (s_1 - s_2)/(s_3 - s_4)$. bEM, sEM, and sEM-vr updates are then defined respectively as Eq. (3), Eq. (4), and Eq. (5).

We construct a dataset of $N = 10,000$ samples drawn from the model with $\mu = 0.5$, and run bEM until convergence (to double precision) to obtain the MLE $\mu_*$. We then measure the convergence of $\mathbb{E} \left\| \hat{\mu}_t - \mu_* \right\|^2$ as well as the variance term $\mathrm{Var}_t[\hat{\mu}_t]/\rho_t^2$ for bEM, sEM, and sEM-vr with respective to the number of epochs. $\mathrm{Var}_t[\hat{\mu}_t]$ is always quadratic with respect to the step size $\rho_t$, so we divide it by $\rho_t^2$ to cancel the effect of the step size, and just study the intrinsic variance. We tune the step size manually, and set $\rho_t = 3/(t + 10)$ for sEM and $\rho = 0.003$ for sEM-vr.

The result is shown as Fig. 1. sEM converges faster than bEM in the first 8 epochs, and then it is outperformed by bEM, because sEM is asymptotically slower, as mentioned in Sec. 2.2. The convergence curve of sEM-vr exhibits a staircase pattern. In the beginning of each epoch it converges very fast because $\|\hat{s}_{e,t} - \hat{s}_{e,0}\|$ is small, so the variance is small. The variance then becomes larger and the convergence slows down. Then we start a new epoch and compute a new $F(\hat{s}_{e,0})$, so that the convergence is fast again. On the other hand, the variance of sEM remains constant.

## 4.2 Probabilistic Latent Semantic Analysis

### 4.2.1 Model and Algorithm

Probabilistic Latent Semantic Analysis (pLSA) [18] represents text documents as mixtures of topics. pLSA takes a list $\mathcal{I}$ of tokens, where each token $i$ is represented by a pair of document and word IDs $(d_i, v_i)$, that indicates for the presence of a word $v_i$ in document $d_i$. Denote $[n] = \{1, \ldots, n\}$, we have $d_i \in [D]$ and $v_i \in [V]$. pLSA assigns a latent topic $z_i \in [K]$ for each token, and defines the joint likelihood as $p(\mathcal{I}, Z|\boldsymbol{\theta}, \boldsymbol{\phi}) = \prod_{i \in \mathcal{I}} \mathrm{Cat}(z_i; \theta_{d_i}) \mathrm{Cat}(v_i; \phi_{z_i})$, with the parameters $\boldsymbol{\theta} = \{\theta_d\}_{d=1}^D$ and $\boldsymbol{\phi} = \{\phi_k\}_{k=1}^K$. We have priors $p(\theta_d) = \mathrm{Dir}(\theta_d; K, \alpha')$ and $p(\phi_k) = \mathrm{Dir}(\phi_k; V, \beta')$, where $\mathrm{Dir}(K, \alpha)$ is a $K$-dimensional symmetric Dirichlet distribution with the concentration parameter $\alpha$, and find an MAP estimation $\mathrm{argmax}_{\boldsymbol{\theta}, \boldsymbol{\phi}} \log \sum_Z p(W, Z|\boldsymbol{\theta}, \boldsymbol{\phi}) + \log p(\boldsymbol{\theta}) + \log p(\boldsymbol{\phi})$. Only the updates are presented here and the derivation is in Appendix D. Let $\gamma_{ik}(\boldsymbol{\theta}, \boldsymbol{\phi}) := p(z_i = k|v_i, \boldsymbol{\theta}, \boldsymbol{\phi}) \propto \theta_{d_i, k} \phi_{k, v_i}$ be the posterior topic assignment of the token $v_i$, bEM updates $\gamma_{dk}(\boldsymbol{\theta}, \boldsymbol{\phi}) = \sum_{i \in \mathcal{I}_d} \gamma_{ik}(\boldsymbol{\theta}, \boldsymbol{\phi})$, and

$\gamma_{kv}(\boldsymbol{\theta}, \boldsymbol{\phi}) = \sum_{i \in \mathcal{I}_v} \gamma_{ik}(\boldsymbol{\theta}, \boldsymbol{\phi})$ in E-step, where $\mathcal{I}_d = \{(d_i, v_i) | d_i = d\}$ and $\mathcal{I}_v = \{(d_i, v_i) | v_i = v\}$. M-step is $\theta_{dk} = (\gamma_{dk} + \alpha)/(\sum_k \gamma_{dk} + K\alpha)$, and $\phi_{kv} = (\gamma_{kv} + \beta)/(\sum_v \gamma_{kv} + V\beta)$, where $\alpha = \alpha' - 1$ and $\beta = \beta' - 1$. We distinguish $(\gamma_{ik}, \gamma_{dk}, \gamma_{vk})$ and $(\mathcal{I}, \mathcal{I}_d, \mathcal{I}_v)$ by indices for simplicity.

sEM approximates the full batch expected sufficient statistics $\gamma_{dk}$ and $\gamma_{kv}$ with exponential moving averages $\hat{s}_{t,d,k}$ and $\hat{s}_{t,k,v}$ at iteration $t$, and updates $\hat{s}_{t+1,d,k} = (1 - \rho_t)\hat{s}_{t,d,k} + \rho_t \frac{|\mathcal{I}|}{|\hat{\mathcal{I}}|} \sum_{i \in \hat{\mathcal{I}}_d} \gamma_{ik}(\hat{\boldsymbol{\theta}}_t, \hat{\boldsymbol{\phi}}_t)$, and $\hat{s}_{t+1,k,v} = (1 - \rho_t)\hat{s}_{t,k,v} + \rho_t \frac{|\mathcal{I}|}{|\hat{\mathcal{I}}|} \sum_{i \in \hat{\mathcal{I}}_v} \gamma_{ik}(\hat{\boldsymbol{\theta}}_t, \hat{\boldsymbol{\phi}}_t)$, where we sample a minibatch $\hat{\mathcal{I}} \subset \mathcal{I}$ of tokens per iteration, $\hat{\mathcal{I}}_d, \hat{\mathcal{I}}_v$ are defined in the same way as $\mathcal{I}_d, \mathcal{I}_v$. $\hat{\boldsymbol{\theta}}_t$ and $\hat{\boldsymbol{\phi}}_t$ are computed in the M-step with $\hat{s}_{t,d,k}$ and $\hat{s}_{t,k,v}$. This sEM algorithm is known as SCVB0 [16].

sEM-vr updates as $\hat{s}_{e,t+1,d,k} = (1 - \rho)\hat{s}_{e,t,d,k} + \rho \frac{|\mathcal{I}|}{|\hat{\mathcal{I}}|} \sum_{i \in \hat{\mathcal{I}}_d} (\gamma_{ik}(\hat{\boldsymbol{\theta}}_{e,t}, \hat{\boldsymbol{\phi}}_{e,t}) - \gamma_{ik}(\hat{\boldsymbol{\theta}}_{e,0}, \hat{\boldsymbol{\phi}}_{e,0})) + \rho\gamma_{dk}(\hat{\boldsymbol{\theta}}_{e,0}, \hat{\boldsymbol{\phi}}_{e,0})$, and $\hat{s}_{e,t+1,k,v} = (1 - \rho)\hat{s}_{e,t,k,v} + \rho \frac{|\mathcal{I}|}{|\hat{\mathcal{I}}|} \sum_{i \in \hat{\mathcal{I}}_v} (\gamma_{ik}(\hat{\boldsymbol{\theta}}_{e,t}, \hat{\boldsymbol{\phi}}_{e,t}) - \gamma_{ik}(\hat{\boldsymbol{\theta}}_{e,0}, \hat{\boldsymbol{\phi}}_{e,0})) + \rho\gamma_{kv}(\hat{\boldsymbol{\theta}}_{e,0}, \hat{\boldsymbol{\phi}}_{e,0})$, where $\gamma_{dk}(\hat{\boldsymbol{\theta}}_{e,0}, \hat{\boldsymbol{\phi}}_{e,0})$ and $\gamma_{kv}(\hat{\boldsymbol{\theta}}_{e,0}, \hat{\boldsymbol{\phi}}_{e,0})$ is computed by bEM per epoch. We have pseudocode for sEM and sEM-vr in Appendix D.

If $\boldsymbol{\theta}$ is integrated out instead of maximized, we recover an MAP estimation [14] of latent Dirichlet allocation (LDA) [4]. Many existing algorithms for LDA actually optimize the pLSA objective as an approximation of the LDA objective, including CVB0 [2, 31, 19], SCVB0 [16], BP-LDA [10], ESCA [37], and WarpLDA [9]. This approximation works well in practice when the number of topics is small [2]. We have more discussions in Appendix D.1.

### 4.2.2 Experimental Settings

We compare sEM-vr with bEM and sEM (SCVB0), which is the start-of-the-art algorithm for pLSA, on four datasets listed in Table 1. We also compare with two gradient based algorithms, stochastic mirror descent (SMD) [10] and reparameterized stochastic gradient descent (RSGD) as well as their variants with SVRG-style [20] variance reduction, denoted as SMD-vr and RSGD-vr, despite their convergence properties are unknown. Both SMD and RSGD replace the M-step with a stochastic gradient step. SMD updates as $\theta_{dk} \propto \theta_{dk} \exp\{\rho\nabla_{\theta_{dk}} Q\}$ and $\phi_{kv} \propto \phi_{kv} \exp\{\rho\nabla_{\phi_{kv}} Q\}$, where $Q$ is defined as Eq. (1). RSGD adopts the reparameterization $\theta_{dk} = \frac{\exp \lambda_{dk}}{\sum_k \exp \lambda_{dk}}$ and $\phi_{kv} = \frac{\exp \tau_{kv}}{\sum_v \exp \tau_{kv}}$, and directly optimize $Q$ w.r.t. $\lambda$ and $\tau$ by stochastic gradient descent. Derivations of SMD and RSGD are in Appendix D.6. All the algorithms are implemented in C++, and are highly-optimized and parallelized. The testing machine has two 12-core Xeon E5-2692v2 CPUs and 64GB main memory.

We assess the convergence of algorithms by the training objective $\log p(W|\theta, \phi) + \log p(\theta|\alpha') + \log p(\phi|\beta')$, i.e., logarithm of unnormalized posterior distribution $p(\theta, \phi|W, \alpha', \beta')$. For each dataset and the number of topics $K \in \{50, 100\}$, we first select the hyperparameters by a grid search $K\alpha \in \{0.1, 1, 10, 100\}$ and $\beta \in \{0.01, 0.1, 1\}$.[3] Then, we do another grid search to choose the step size. For sEM-vr, we choose $\rho \in \{0.01, 0.02, 0.05, 0.1, 0.2\}$, and for all other stochastic algorithms, we set $\rho_t = a/(t + t_0)^\kappa$, and choose $a \in \{10^{-7}, \ldots, 10^0\}, t_0 \in \{10, 100, 1000\}$ and $\kappa \in \{0.5, 0.75, 1\}$.[4] Finally, we repeat 5 runs with difference random seeds for each algorithm with its best step size. $E$ is 20 for NIPS and NYTimes, and 5 for Wiki and PubMed. $M$ is 50 for NIPS and 500 for all the other datasets.

### 4.2.3 Results for pLSA

We plot the training objective against running time as first and second row of Fig. 2. We find that gradient-based algorithms and bEM are not competitive with sEM and sEM-vr, so we only report their results on NIPS, to make the distinction sEM and sEM-vr more clear. Full results and more explanations of the slow convergence of gradient-based algorithms are available in Appendix D.6. Due to the reduced variance, sEM-vr consistently converges faster to better training objective than sEM and bEM on all the datasets, while the constant step size of sEM-vr is easier to tune than the decreasing sequence of step sizes for sEM.

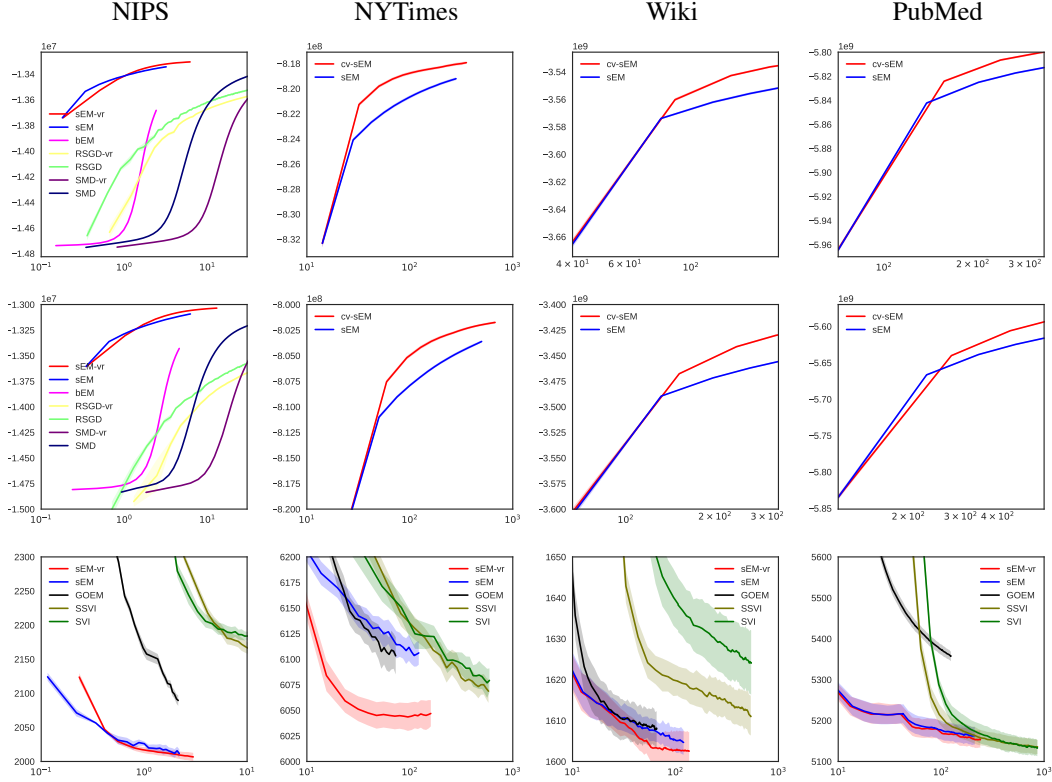

Figure 2: pLSA and LDA convergence results. X-axis is running time in seconds. First and second row: pLSA with $K = 50$ and $K = 100$, y-axis is the training objective. Third row: LDA with $K = 10$, y-axis is the testing perplexity.

## 4.3 Results for LDA

As discussed in Sec. 4.2.1, algorithms for pLSA also work well as approximate training algorithms for LDA, if the number of topics is small. Therefore, we also evaluate our sEM-vr algorithm for LDA, with a small number of $K = 10$ topics. The training algorithm is exactly the same, but the evaluation metric is different. We hold out a small testing set, and report the testing perplexity, computed by the left-to-right algorithm [34] on the testing set. We compare with a state-of-the-art algorithm, Gibbs online expectation maximization (GOEM) [14], which outperforms a wide range of algorithms including SVI [17], hybrid variational-Gibbs [27], and SGRLD [28]. We also compare with stochastic variational inference (SVI) [17] and its variance reduced variant SSVI [25].

The third row of Fig. 2 shows the results. We observed that sEM-vr converges the fastest on all the datasets except NIPS, where sEM converges faster due to its cheaper iterations. sEM-vr always gets better results than sEM in the end. GOEM converges slower due to its high Monte-Carlo variance. SVI and SSVI converge slower due to their inexact mean field assumption and expensive iterations, including an inner loop for inferring the local latent variables and frequent evaluation of the expensive digamma function. For a larger number of topics, such as 100, we find that GOEM performs the best since it does not approximate LDA as pLSA, and does not make mean field assumptions as SVI and SSVI. Extending our algorithm to variational EM and Monte-Carlo EM, when the E-step is not tractable, is an interesting future direction.

## 5 Conclusions and Discussions

We propose a variance reduced stochastic EM (sEM-vr) algorithm. sEM-vr achieves a $\left(1 + \log(M/\kappa^2)\right)^{-E}$ local convergence rate, which is faster than both the $(1 - \lambda)^{-2E}$ rate of batch EM and $O(T^{-1})$ rate of plain stochastic EM (sEM). Unlike sEM, which requires a decreasing sequence of step sizes to converge, sEM-vr only requires a constant step size to achieve this local

convergence rate as well as global convergence, under stronger assumptions. We compare sEM-vr against bEM, sEM and other gradient and Bayesian algorithms, on GMM and pLSA tasks, and find that sEM-vr converges significantly faster than these alternatives.

An interesting future direction is leveraging recent progress on variance reduced stochastic gradient descent for non-convex optimization [23] to relax our assumptions on strongly-log-concavity, and extend sEM-vr to stochastic control variates, which works better on very large data sets. Extending our work to variational EM and Monte-Carlo EM is also interesting.

**Acknowledgments**

We thank Chris Maddison, Adam Foster, and Jin Xu for proofreading. J.C. and J.Z. were supported by the National Key Research and Development Program of China (No.2017YFA0700904), NSFC projects (Nos. 61620106010, 61621136008, 61332007), the MIIT Grant of Int. Man. Comp. Stan (No. 2016ZXFB00001), Tsinghua Tiangong Institute for Intelligent Computing, the NVIDIA NVAIL Program and a Project from Siemens. YWT was supported by funding from the European Research Council under the European Union's Seventh Framework Programme (FP7/2007-2013) ERC grant agreement no. 617071, and from Tencent AI Lab through the Oxford-Tencent Collaboration on Large Scale Machine Learning.

## Footnotes

[2] Without affecting the convergence rates, we slightly adjust the convergence theorems in [6, 12] to view them in a uniformed way, see Appendix A for details.

[3] We find that all the algorithms have the same best hyperparameter configuration.

[4] We have tried constant step sizes for SMD-vr and RSGD-vr but found it worse than decreasing step sizes.

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
