[Supplementary Material · supplementary.pdf]

# A Some Clarifications

## A.1 Convergence rate of bEM

Dempster et al. [11] showed in their Theorem 4 that the convergence rate of bEM is

$$\left\|\hat{\theta}_E - \theta_*\right\|^2 \leq (1-\lambda)^{-2E}\left\|\hat{\theta}_0 - \theta_*\right\|^2,$$

where $1 - \lambda$ is the maximum eigenvalue of $\partial R(F(\theta_*))/\partial\theta_*$. We define $1 - \lambda$ in Sec. 3.3 as the maximum eigenvalue of $\partial F(s_*)/\partial s_*$. The two definitions are equivalent because at the stationary point $(\theta_*, s_*)$, we have $\theta_* = R(s_*)$ and $s_* = F(\theta_*)$. Note that for two matrices $A$ and $B$, $AB$ and $BA$ have the same spectrum. Therefore, $\partial F(s_*)/\partial s_* = \partial F(R(s_*))/\partial s_* = (\partial F/\partial\theta_*)(\partial R/\partial s_*)$ has the same spectrum with $\partial R(F(\theta_*))/\partial\theta_* = (\partial R/\partial s_*)(\partial F/\partial\theta_*)$, so $1 - \lambda$ is the maximum eigenvalue of both $\partial F(s_*)/\partial s_*$ and $\partial R(F(\theta_*))/\partial\theta_*$.

Dempster et. al [11] also showed that $\partial R(F(\theta_*))/\partial\theta_* = (I_* - \nabla^2\mathcal{L}(\theta_*))^{-1}I_*$, where $I_* = -\mathbb{E}_{p(H|X,\theta_*)}\nabla^2\log p(H|X,\theta_*) \succeq 0$ is the Fisher information of $p(H|X,\theta_*)$, $A \succeq B$ means $A - B$ is positive semidefinite, and $A \succ B$ means $A - B$ is positive definite. If $0 \succ \nabla^2\mathcal{L}(\theta_*)$, as we assumed in Theorem 1, then $I_* - \nabla^2\mathcal{L}(\theta_*) \succ I_* \succeq 0$, and the eigenvalues of $\partial R(F(\theta_*))/\partial\theta_*$ are between $[0, 1)$, so $\lambda > 0$.

Finally, the convergence of the sequence of parameters $(\hat{\theta}_t)$ and sufficient statistics $(\hat{s}_t)$ are equivalent as long as the mappings between them, $R(s)$ and $F(\theta)$, are Lipschitz continuous.

## A.2 Convergence rate of sEM

Cappe and Moulines [5] showed in their Theorem 2 that the sequence $\rho_T^{-1/2}(\hat{\theta}_T - \theta_*)$ converge in distribution to $\mathcal{N}(0, \Sigma(\theta_*))$, where $\Sigma(\theta_*)$ is irrelevant with $\rho_T$. This implies $\rho_T^{-1}\left\|\hat{\theta}_T - \theta_*\right\|^2 \to \Sigma(\theta_*)$, that is $\left\|\hat{\theta}_T - \theta_*\right\|^2 = O(\rho_T)$. Finally, we convert the convergence of $(\hat{\theta}_t)$ to the convergence of $(\hat{s}_t)$ as mentioned in Sec. A.1.

# B Remaining Proof of Theorem 1

*Proof.* We continue the analysis in Sec. 3.3 of the sequence $a_{t+1} \leq (1-\epsilon\rho)a_t + c\rho^2 a_0$, where $a_t = \mathbb{E}\left\|\Delta_t\right\|^2$, $\epsilon = \lambda/4$ and $c = 8L_f^2$. We have

$$\begin{aligned}
a_M &\leq (1-\epsilon\rho)a_{M-1} + c\rho^2 a_0 \\
&\leq (1-\epsilon\rho)^M a_0 + c\rho^2\left[1 + (1-\epsilon\rho) + \cdots + (1-\epsilon\rho)^{M-1}\right]a_0 \\
&\leq \exp(-M\epsilon\rho)a_0 + c\rho^2\frac{1-(1-\epsilon\rho)^M}{\epsilon\rho}a_0 \\
&\leq \left[\exp(-M\epsilon\rho) + \frac{c\rho}{\epsilon}\right]a_0 := A_M,
\end{aligned}$$ (11)

where the third line utilizes the inequality $1 + x \leq \exp(x), \forall x \in \mathbb{R}$. Taking derivative of the upper bound $A_M$ w.r.t. $\rho$, we have

$$(A_M)'_\rho = \left[-M\epsilon\exp(-M\epsilon\rho) + \frac{c}{\epsilon}\right]a_0.$$

Let the derivative be zero, we obtain the optimal upper bound and its corresponding $\rho$, denoted as $\rho_*$

$$\rho_* = \log\left(\frac{\epsilon^2 M}{c}\right)/(\epsilon M),$$ (12)

$$a_M \leq \frac{c}{\epsilon^2 M}\left(1 + \log\frac{\epsilon^2 M}{c}\right)a_0.$$ (13)

Plugging $a_t = \mathbb{E} \left\| \Delta_t \right\|^2$, $\epsilon = \lambda/4$ and $c = 8L_f^2$ into Eq. (11, 12, 13), we have

$$\mathbb{E} \left\| \Delta_M \right\|^2 \leq \left[ \exp\left(-M\lambda\rho/4\right) + 32L_f^2\rho/\lambda \right] \left\| \Delta_0 \right\|^2$$

$$\rho_* = 4\log(M/\kappa^2)/(\lambda M) = 4\log\left(\lambda^2 M/(128L_f^2)\right)/(\lambda M)$$

$$\mathbb{E} \left\| \Delta_M \right\|^2 \leq \left[ \left(1 + \log(M/\kappa^2)\right) \kappa^2/M \right] \left\| \Delta_0 \right\|^2,$$

where $\kappa^2 = \frac{c}{\epsilon^2} = \frac{128L_f^2}{\lambda^2}$. We can verify that $\rho_* = 4\log\left(\lambda^2 M/(128L_f^2)\right)/(\lambda M)$ is less equal than $\lambda/(32L_f^2)$, assumed by Theorem 1 because $\log x < x$ for all $x > 0$, where $x = \lambda^2 M/(128L_f^2)$.

Finally, because we take $\hat{s}_{E+1,0} = \hat{s}_{E,M}$, we get Eq. (6, 7). $\qquad\square$

## C  Proof of Theorem 2

We construct an auxiliary function

$$\hat{Q}_{e,t}(\theta) = N\left(\eta(\theta)^\top \hat{s}_{e,t} - A(\theta)\right),$$

and its equivalent recursive definition

$$\hat{Q}_{e,t+1}(\theta) = (1-\rho)\hat{Q}_{e,t}(\theta) + \rho(Q_i(\theta; \hat{\theta}_{e,t}) - Q_i(\theta; \hat{\theta}_{e,0}) + Q(\theta; \hat{\theta}_{e,0})),$$

$$\hat{Q}_{0,0}(\theta) = Q(\theta; \hat{\theta}_{0,0}),$$

where $Q(\theta; \hat{\theta}_{e,0})$ is defined in Eq. (1), $Q_i(\theta; \hat{\theta}_{e,0}) = \mathbb{E}_{p(h_i|x_i,\hat{\theta}_{e,0})}[\log p(x_i, h_i; \theta)] = \eta(\theta)^\top f_i(\hat{\theta}_{e,0}) - A(\theta)$, and $\hat{\theta}_{e,t} := \mathrm{argmax}_\theta \hat{Q}_{e,t}(\theta) = R(\hat{s}_{e,t})$. This is similar to the original from of sEM [5] rather than its exponential family form we present in the main text.

According to Assumption (b), $\log p(x_i, h_i; \theta)$ is $\gamma$-strongly-concave, so $Q_i(\theta; \hat{\theta}) = \mathbb{E}_{p(h_i|x_i,\hat{\theta})}[\log p(x_i, h_i; \theta)]$ is also $\gamma$-strongly-concave with respect to $\theta$. By induction, $\mathbb{E}_{e,t}[\hat{Q}_{e,t+1}(\theta)] = (1-\rho)\hat{Q}_{e,t}(\theta) + \rho Q(\theta; \hat{\theta}_{e,t})$ is also $\gamma$-strongly-concave for all $e$ and $t$.

By the recursive formulation, we have

$$Q(\hat{\theta}_{e,t+1}; \hat{\theta}_{e,0}) - Q(\hat{\theta}_{e,t}; \hat{\theta}_{e,0}) = \frac{1}{\rho}\left(\hat{Q}_{e,t+1}(\hat{\theta}_{e,t+1}) - \hat{Q}_{e,t+1}(\hat{\theta}_{e,t})\right) + \frac{1-\rho}{\rho}\left(\hat{Q}_{e,t}(\hat{\theta}_{e,t}) - \hat{Q}_{e,t}(\hat{\theta}_{e,t+1})\right)$$

$$+ Q_i(\theta_{e,t}; \hat{\theta}_{e,t}) - Q_i(\theta_{e,t+1}; \hat{\theta}_{e,t}) + Q_i(\theta_{e,t+1}; \hat{\theta}_{e,0}) - Q_i(\theta_{e,t}; \hat{\theta}_{e,0}).$$

According to the definition of $\hat{\theta}_{e,t}$, and assuming that the algorithm has not converged, we have

$$\hat{Q}_{e,t+1}(\hat{\theta}_{e,t+1}) - \hat{Q}_{e,t+1}(\hat{\theta}_{e,t}) > 0,$$

$$\hat{Q}_{e,t}(\hat{\theta}_{e,t}) - \hat{Q}_{e,t}(\hat{\theta}_{e,t+1}) > 0,$$

Moreover,

$$\mathbb{E}_{e,t}[Q_i(\hat{\theta}_{e,t}; \hat{\theta}_{e,t}) - Q_i(\hat{\theta}_{e,t+1}; \hat{\theta}_{e,t}) + Q_i(\hat{\theta}_{e,t+1}; \hat{\theta}_{e,0}) - Q_i(\hat{\theta}_{e,t}; \hat{\theta}_{e,0})].$$

$$= \mathbb{E}_{e,t}[\eta(\hat{\theta}_{e,t})^\top f_i(\hat{\theta}_{e,t}) - \eta(\hat{\theta}_{e,t+1})^\top f_i(\hat{\theta}_{e,t}) + \eta(\hat{\theta}_{e,t+1})^\top f_i(\hat{\theta}_{e,0}) - \eta(\hat{\theta}_{e,t})^\top f_i(\hat{\theta}_{e,0})]$$

$$= \left(\eta(\hat{\theta}_{e,t}) - \eta(\hat{\theta}_{e,t+1})\right)^\top \left(F(\hat{\theta}_{e,t}) - F(\hat{\theta}_{e,0})\right).$$

Therefore,

$$\mathbb{E}_{e,t}[Q(\hat{\theta}_{e,t+1}; \hat{\theta}_{e,0}) - Q(\hat{\theta}_{e,t}; \hat{\theta}_{e,0})]$$

$$> \frac{1}{\rho}\left(\hat{Q}_{e,t+1}(\hat{\theta}_{e,t+1}) - \hat{Q}_{e,t+1}(\hat{\theta}_{e,t})\right) + \left(\eta(\hat{\theta}_{e,t}) - \eta(\hat{\theta}_{e,t+1})\right)^\top \left(F(\hat{\theta}_{e,t}) - F(\hat{\theta}_{e,0})\right)$$

$$\geq \frac{\gamma}{2\rho}\left\| \hat{\theta}_{e,t+1} - \hat{\theta}_{e,t} \right\|^2 - L_\eta L_f \left\| \hat{\theta}_{e,t} - \hat{\theta}_{e,t+1} \right\| \left\| \hat{\theta}_{e,t} - \hat{\theta}_{e,0} \right\|, \tag{14}$$

where the last line utilizes the $\gamma$-strong-concavity of $\hat{Q}_{e,t+1}$ (recall that $\nabla \hat{Q}_{e,t+1}(\hat{\theta}_{e,t+1}) = 0$, according to the definition of $\hat{\theta}_{e,t+1}$) as well as Lipschitz continuity of $\eta$ and $f_i$. Summing up Eq. (14), we have

$$\mathbb{E}[Q(\hat{\theta}_{e,M}; \hat{\theta}_{e,0}) - Q(\hat{\theta}_{e,0}; \hat{\theta}_{e,0})]$$

$$> \frac{\gamma}{2\rho} \sum_{t=0}^{M-1} \left\| \hat{\theta}_{e,t+1} - \hat{\theta}_{e,t} \right\|^2 - L_\eta L_f \sum_{t=0}^{M-1} \left\| \hat{\theta}_{e,t} - \hat{\theta}_{e,t+1} \right\| \left\| \hat{\theta}_{e,t} - \hat{\theta}_{e,0} \right\|$$

$$\geq \frac{\gamma}{2\rho} \Delta_e^2 - M(M-1)L_\eta L_f \Delta_e^2/2,$$

where $\Delta_e := \max_t \left\| \hat{\theta}_{e,t+1} - \hat{\theta}_{e,t} \right\|$. Therefore, when $\rho < \frac{\gamma}{M(M-1)L_\eta L_f}$, we have $\mathbb{E}[Q(\hat{\theta}_{e,M}; \hat{\theta}_{e,0}) - Q(\hat{\theta}_{e,0}; \hat{\theta}_{e,0})] > 0$ for any $\hat{\theta}_{e,0}$ and $\hat{\theta}_{e,M}$. That is, sEM-vr improves the lower bound of the log marginal likelihood $\mathcal{L}$ in each epoch. Hence sEM-vr can be considered as a generalized EM (GEM) algorithm [34], which improves the ELBO in every epoch. Applying Wu's Theorem 1 [34], we conclude that sEM-vr converges globally to a stationary point.

## D Details of Probabilistic Latent Semantic Analysis

In pLSA, we want to model a collection of $D$ documents $W = \{w_d\}_{d=1}^D$, where each document $w_d = \{w_{dn}\}_{n=1}^{N_d}$ is a list of tokens, and each token $w_{dn} \in \{1, \ldots, V\}$ is represented by its ID in a vocabulary of $V$ words. The notations here is different with the main text, but rather similar with the SCVB0 paper [15].

We define the following generative procedure of the documents:

1. for each topic $k \in [K]$, generate $\phi_k \sim \text{Dir}(V, \beta')$;
2. for each document $d \in [D]$, generate $\theta_d \sim \text{Dir}(K, \alpha')$;
3. for each position $d \in [D], n \in [N_d]$, generate $z_{dn} \sim \text{Cat}(\theta_d)$, generate $w_{dn} \sim \text{Cat}(\phi_{z_{dn}})$,

where $[K] := \{1, \ldots, K\}$, $\text{Dir}(K, \alpha)$ is a $K$-dimensional symmetric Dirichlet distribution with the concentration parameter $\alpha$, and $\text{Cat}(\cdot)$ is a categorical distribution. This is exactly the same generative procedure with latent Dirichlet allocation (LDA) [3].

Denote $Z$, $\theta$ and $\phi$ to be the collection of $z_{dn}$, $\theta_d$ and $\phi_k$, the generative procedure defines a joint distribution $p(W, Z, \theta, \phi | \alpha', \beta')$. Our goal is a maximum *a posteriori* (MAP) estimate of the parameters $(\theta, \phi)$.

$$\underset{\theta, \phi}{\arg\max} \log p(\theta, \phi | W, \alpha', \beta')$$

$$= \underset{\theta, \phi}{\arg\max} \log \sum_Z \{p(W, Z | \theta, \phi) p(\theta | \alpha') p(\phi | \beta')\}. \tag{15}$$

Let $\alpha = \alpha' - 1$ and $\beta = \beta' - 1$, we have

$$p(W, Z | \theta, \phi) p(\theta | \alpha') p(\phi | \beta') \propto \prod_{dn} \theta_{d, z_{dn}} \phi_{z_{dn}, w_{dn}} \prod_{dk} \theta_{dk}^\alpha \prod_{kv} \phi_{kv}^\beta.$$

$$= \prod_{dk} \theta_{dk}^{C_{dk} + \alpha} \prod_{kv} \phi_{kv}^{C_{kv} + \beta}$$

$$= \exp \left\{ \sum_{dk} (C_{dk} + \alpha) \log \theta_{dk} + \sum_{kv} (C_{kv} + \beta) \log \phi_{kv} \right\}, \tag{16}$$

where $C_{dk} = \sum_n \mathbb{I}(z_{dn} = k)$ and $C_{kv} = \sum_{dn} \mathbb{I}(z_{dn} = k)\mathbb{I}(w_{dn} = v)$, and $\mathbb{I}(\cdot)$ is the indicator function. Eq. (16) is in an exponential family form where $(C_{dk}, C_{kv})$ are the sufficient statistics, and $(\log \theta_{dk}, \log \phi_{kv})$ are the natural parameters.

Then,

$$Q(\theta, \phi; \theta', \phi') = \mathbb{E}_{p(Z | W, \theta', \phi')} [\log p(W, Z | \theta, \phi)] + \log p(\theta | \alpha') + \log p(\phi | \beta') + \text{const.}$$

$$= \sum_{dk} (\gamma_{dk}(\theta', \phi') + \alpha) \log \theta_{dk} + \sum_{kv} (\gamma_{kv}(\theta', \phi') + \beta) \log \phi_{kv}, \tag{17}$$

where const. is a constant term w.r.t. $\theta$ and $\phi$,

$$\gamma_{dnk}(\theta, \phi) = \mathbb{E}_{p(z_{dn}|w_{dn}, \theta, \phi)}[\mathbb{I}(z_{dn} = k)] = p(z_{dn} = k|w_{dn}, \theta, \phi) = \frac{\theta_{dk}\phi_{k,w_{dn}}}{\sum_k \theta_{dk}\phi_{k,w_{dn}}}.$$

and

$$\gamma_{dk}(\theta, \phi) := \mathbb{E}_{p(Z|W,\theta,\phi)}[C_{dk}] = \sum_n \gamma_{dnk}(\theta, \phi), \tag{18}$$

$$\gamma_{kv}(\theta, \phi) := \mathbb{E}_{p(Z|W,\theta,\phi)}[C_{kv}] = \sum_{dn} \mathbb{I}(w_{dn} = v)\gamma_{dnk}(\theta, \phi). \tag{19}$$

## D.1 Connection with LDA

The only difference of our pLSA objective Eq. (15) with LDA [3] is whether treating $\theta$ as latent variable or parameter. If $\theta$ is marginalized out, we recover the LDA objective

$$\operatorname*{argmax}_{\phi} \log p(\phi|W, \alpha', \beta')$$

$$= \operatorname*{argmax}_{\phi} \log \sum_Z \int_\theta \{p(W, Z|\theta, \phi)p(\theta|\alpha')p(\phi|\beta')\} \, d\theta. \tag{20}$$

Due to their resemblance, a number of LDA training algorithms optimizes the pLSA training objective Eq. (15) instead of the LDA training objective Eq. (20) for faster convergence, including CVB0 [2, 30, 18], SCVB0 [15], BP-LDA [9], ESCA [35], and WarpLDA [8]. This approximation works well in practice [2] when the number of topics is small.

## D.2 E-step

In the E-step, we compute the expected sufficient statistics $\gamma_{dk}(\theta, \phi)$ and $\gamma_{kv}(\theta, \phi)$ as Eq. (18, 19).

## D.3 M-step

In the M-step, we solve the maximization problem

$$\operatorname*{argmax}_{\theta, \phi} \sum_{dk} (\gamma_{dk} + \alpha) \log \theta_{dk} + \sum_{kv} (\gamma_{kv} + \beta) \log \phi_{kv}. \tag{21}$$

$$\text{s.t. } \sum_k \theta_{dk} = 1, \forall d \in [D],$$

$$\sum_v \phi_{kv} = 1, \forall k \in [K].$$

The solution is

$$\theta_{dk} = \frac{\gamma_{dk} + \alpha}{\sum_k \gamma_{dk} + K\alpha}, \quad \phi_{kv} = \frac{\gamma_{kv} + \beta}{\sum_v \gamma_{kv} + V\beta}.$$

## D.4 Stochastic EM Updates

According to Sec. 2.2, we can derive an sEM algorithm by replacing the E-step with stochastic approximation. sEM algorithm for pLSA is known as SCVB0 [15]. SCVB0 optimizes $\theta$ and $\phi$ alternatively. To optimize $\theta_d$ for a document $d$ given $\phi$, SCVB0 replaces $\gamma_{dk}$, the sum over all the tokens $n \in [N_d]$ (Eq. 18), with a stochastic approximation

$$\text{E step: } \hat{s}_{t+1,d,k} = (1 - \rho_t)\hat{s}_{t,d,k} + \rho_t N_d \gamma_{dnk}(\theta_t, \phi), \quad n \sim \text{Uniform}(N_d),$$

$$\text{M step: } \theta_{t+1,d,k} = (\hat{s}_{t+1,d,k} + \alpha)/(\sum_k \hat{s}_{t+1,d,k} + K\alpha),$$

where $\hat{s}_{t,d,k}$ is an approximation of the batch sufficient statistics $\gamma_{dk}$, and $\theta_{t,d,k}$ is the estimated parameter at iteration $t$.

---
**Algorithm 1** Batch E-step for PLSA.

---
**Require:** $\theta, \phi, W$
  $\forall d, k, \gamma_{dk} \leftarrow 0$
  $\forall k, v, \gamma_{kv} \leftarrow 0$
  **for** each document $d$ **do**
    **for** each token $w_{dn}$ **do**
      $\forall k, \gamma_{dnk} = \theta_{dk}\phi_{k,w_{dn}}/(\sum_k \theta_{dk}\phi_{k,w_{dn}})$
      $\forall k, \gamma_{dk} \leftarrow \gamma_{dk} + \gamma_{dnk}, \gamma_{k,w_{dn}} \leftarrow \gamma_{k,w_{dn}} + \gamma_{dnk}$
    **end for**
  **end for**
  Return $\gamma_{dk}, \gamma_{kv}$.

---

---
**Algorithm 2** SCVB0 algorithm for PLSA.

---
**Require:** Initial $\theta, \phi$
  $\hat{s}_{d,k}, \hat{s}_{k,v} \leftarrow \text{BatchEStep}(\theta, \phi, W)$ (Alg. 1)
  **for** each minibatch of $M$ documents **do**
    Compute the step size $\rho$
    (Update $\theta$)
    **for** each document $d$ **do**
      **for** each token $w_{dn}$ **do**
        Compute $\forall k, \gamma_{dnk} = \theta_{dk}\phi_{k,w_{dn}}/(\sum_k \theta_{dk}\phi_{k,w_{dn}})$,
        E-step: $\forall k, \hat{s}_{d,k} \leftarrow (1-\rho)\hat{s}_{d,k} + \rho N_d \gamma_{dnk}$
        M-step: $\forall k, \theta_{dk} \leftarrow (\hat{s}_{d,k} + \alpha)/(N_d + K\alpha)$.
      **end for**
    **end for**
    (Update $\phi$)
    $\forall k, v, \hat{s}_{kv} \leftarrow (1-\rho)\hat{s}_{kv}$
    **for** each document $d$ **do**
      **for** each token $w_{dn}$ **do**
        Compute $\forall k, \gamma_{dnk} = \theta_{dk}\phi_{k,w_{dn}}/(\sum_k \theta_{dk}\phi_{k,w_{dn}})$,
        E-step: $\forall k, \hat{s}_{k,w_{dn}} \leftarrow \hat{s}_{k,w_{dn}} + \rho\frac{D}{M}\gamma_{dnk}$
      **end for**
    **end for**
    M-step: $\forall k, v, \phi_{kv} \leftarrow (\hat{s}_{kv} + \beta)/(\sum_v \hat{s}_{kv} + V\beta)$.
  **end for**

---

To optimize $\phi$ given $\theta$, SCVB0 randomly sample a minibatch $\mathcal{D} = \{d_1, \ldots, d_M\}$ of $M$ documents, and approximate the sum over the entire corpus, $\gamma_{kv}$, with $\hat{s}_{t,k,v}$

$$\text{E step: } \hat{s}_{t+1,k,v} = (1-\rho_t)\hat{s}_{t,k,v} + \rho_t \frac{D}{M} \sum_{d \in \mathcal{D}} \sum_n \mathbb{I}(w_{dn} = v)\gamma_{dnk}(\theta, \phi_t),$$

$$\text{M step: } \phi_{t+1,k,v} = (\hat{s}_{t+1,k,v} + \beta)/(\sum_v \hat{s}_{t+1,k,v} + V\beta),$$

where $\phi_t$ is the estimated $\phi$ at iteration $t$. See Alg. 2 for a pseudocode.

### D.5 Stochastic EM with Variance Reduction

At each epoch $e$, sEM-vr computes the full-batch sufficient statistics $\gamma_{dk}(\theta_{e,0}, \phi_{e,0})$ and $\gamma_{kv}(\theta_{e,0}, \phi_{e,0})$ according to Eq. (18, 19), and performs the following E-step updates:

$$\hat{s}_{e,t+1,d,k} = (1-\rho)\hat{s}_{e,t,d,k} + \rho\left(N_d\gamma_{dnk}(\theta_{e,t}, \phi_{e,t}) - N_d\gamma_{dnk}(\theta_{e,0}, \phi_{e,0}) + \gamma_{dk}(\theta_{e,0}, \phi_{e,0})\right),$$

$$\hat{s}_{e,t+1,k,v} = (1-\rho)\hat{s}_{e,t,k,v} + \rho\left(\frac{D}{M}\sum_{d \in \mathcal{D}}\sum_n \mathbb{I}(w_{dn} = v)\left(\gamma_{dnk}(\theta_{e,t}, \phi_{e,t}) - \gamma_{dnk}(\theta_{e,0}, \phi_{e,0})\right) + \gamma_{kv}(\theta_{e,0}, \phi_{e,0})\right),$$

see Alg. 3 for the pseudocode.

---

**Algorithm 3** sEM-vr for PLSA.

---
**Require:** Initial $\theta, \phi$
$\quad \hat{s}_{d,k}, \hat{s}_{k,v} \leftarrow \text{BatchEStep}(\theta, \phi, W)$ (Alg. 1)
$\quad$ **for** each epoch $e$ **do**
$\qquad$ Store $\forall d, k, \tilde{\theta}_{d,k} \leftarrow \hat{\theta}_{d,k}, \forall k, v, \tilde{\phi}_{k,v} \leftarrow \hat{\phi}_{k,v}$
$\qquad \tilde{s}_{d,k}, \tilde{s}_{k,v} \leftarrow \text{BatchEStep}(\theta, \phi, W)$ (Alg. 1)
$\qquad$ **for** each minibatch of $M$ documents **do**
$\qquad\qquad$ (Update $\theta$)
$\qquad\qquad$ **for** each document $d$ **do**
$\qquad\qquad\qquad$ **for** each token $w_{dn}$ **do**
$\qquad\qquad\qquad\qquad$ Compute $\forall k, \gamma_{dnk} = \theta_{dk}\phi_{k,w_{dn}}/(\sum_k \theta_{dk}\phi_{k,w_{dn}})$,
$\qquad\qquad\qquad\qquad$ Compute $\forall k, \tilde{\gamma}_{dnk} = \tilde{\theta}_{dk}\tilde{\phi}_{k,w_{dn}}/(\sum_k \tilde{\theta}_{dk}\tilde{\phi}_{k,w_{dn}})$,
$\qquad\qquad\qquad\qquad$ E-step: $\forall k, \hat{s}_{d,k} \leftarrow (1-\rho)\hat{s}_{d,k} + \rho(N_d\gamma_{dnk} - N_d\tilde{\gamma}_{dnk} + \tilde{\gamma}_{dk})$
$\qquad\qquad\qquad\qquad$ M-step: $\theta_d \leftarrow \text{Proj}(\hat{s}_d, \alpha, K)$.
$\qquad\qquad\qquad$ **end for**
$\qquad\qquad$ **end for**
$\qquad\qquad$ (Update $\phi$)
$\qquad\qquad \forall k, v, \hat{s}_{kv} \leftarrow (1-\rho)\hat{s}_{kv} + \rho\tilde{\gamma}_{kv}$
$\qquad\qquad$ **for** each document $d$ **do**
$\qquad\qquad\qquad$ **for** each token $w_{dn}$ **do**
$\qquad\qquad\qquad\qquad$ Compute $\forall k, \gamma_{dnk} = \theta_{dk}\phi_{k,w_{dn}}/(\sum_k \theta_{dk}\phi_{k,w_{dn}})$,
$\qquad\qquad\qquad\qquad$ Compute $\forall k, \tilde{\gamma}_{dnk} = \tilde{\theta}_{dk}\tilde{\phi}_{k,w_{dn}}/(\sum_k \tilde{\theta}_{dk}\tilde{\phi}_{k,w_{dn}})$,
$\qquad\qquad\qquad\qquad$ E-step: $\forall k, \hat{s}_{k,w_{dn}} \leftarrow \hat{s}_{k,w_{dn}} + \rho(\frac{D}{M}\gamma_{dnk} - \frac{D}{M}\tilde{\gamma}_{dnk})$
$\qquad\qquad\qquad$ **end for**
$\qquad\qquad$ **end for**
$\qquad\qquad$ M-step: $\phi_k \leftarrow \text{Proj}(\hat{s}_k, \beta, V)$.
$\qquad$ **end for**
$\quad$ **end for**

---

A subtlety here is $\hat{s}_{e,t,d,k}$ and $\hat{s}_{e,t,k,v}$ can be negative, so we need additional constraints to ensure that $\theta_{dk}$ and $\phi_{kv}$ are non-negative. We solve the following problem in M-step instead of Eq. (21).

$$\operatorname*{argmax}_{\theta,\phi} \sum_{dk}(\gamma_{dk} + \alpha)\log\theta_{dk} + \sum_{kv}(\gamma_{kv} + \beta)\log\phi_{kv}.$$

$$\text{s.t. } \sum_k \theta_{dk} = 1, \forall d \in [D],$$

$$\sum_v \phi_{kv} = 1, \forall k \in [K].$$

$$\theta_{dk} > \epsilon, \forall d \in [D], k \in [K]$$

$$\phi_{kv} > \epsilon, \forall k \in [K], v \in [V],$$

where $\epsilon > 0$ is a threshold to avoid numerical problems. We adopt $\epsilon = 10^{-10}$ in all our experiments. The solution is

$$\theta_d = \text{Proj}(\gamma_d, \alpha, K), \quad \phi_k = \text{Proj}(\gamma_k, \beta, V),$$

where

$$\text{Proj}(\gamma_d, \alpha, K)_k = \epsilon + (1 - K\epsilon)[\gamma_{dk} + \alpha]_+ / \sum_k [\gamma_{dk} + \alpha]_+, \quad [a]_+ := \max\{a, 0\}.$$

### D.6 Gradient-based Updates

Instead of performing exact maximization of the ELBO in the M-step, we can also do a stochastic gradient step. However, as the parameters $\theta_d$ and $\phi_k$ are on probabilistic simplex, i.e., $\theta_{dk} > 0$, $\sum_k \theta_{dk} = 1, \phi_{kv} > 0$ and $\sum_v \phi_{kv} = 0$, standard stochastic gradient descent (SGD) for unconstrained minimization is not applicable. We implement two algorithms, stochastic mirror descent (SMD) [9] and reparameterized SGD (RSGD), for minimizing on the simplex.

SMD update parameters by $\theta_{dk} \propto \theta_{dk} \exp(\rho \nabla_{\theta_{dk}} Q)$ and $\phi_{kv} \propto \phi_{kv} \exp(\rho \nabla_{\phi_{kv}} Q)$, where $Q$ is the ELBO defined as Eq. (17). The updates are

$$\theta_{dk} \propto \theta_{dk} \exp\left(\rho(N_d \gamma_{dnk} + \alpha)/\theta_{dk}\right),$$

$$\phi_{kv} \propto \phi_{kv} \exp\left[\rho\left(\beta + \frac{D}{M}\sum_{d\in\mathcal{D}}\sum_n \mathbb{I}(w_{dn} = v)\gamma_{dnk}\right)/\phi_{kv}\right].$$

RSGD applies the reparametrization

$$\theta_{dk} = \frac{\exp\lambda_{dk}}{\sum_k \exp\lambda_{dk}}, \quad \phi_{kv} = \frac{\exp\tau_{kv}}{\sum_v \exp\tau_{kv}},$$

and optimizes the reparameterized ELBO

$$Q(\lambda, \tau; \theta, \phi) = \sum_{dk}\left(\gamma_{dk}(\theta, \phi) + \alpha\right)\lambda_{dk} - \sum_d (N_d + K\alpha)\log\left(\sum_k \exp\lambda_{dk}\right)$$

$$+ \sum_{kv}\left(\gamma_{kv}(\theta, \phi) + \beta\right)\tau_{kv} - \sum_k\left(\sum_v \gamma_{kv}(\theta, \phi) + V\beta\right)\log\left(\sum_v \exp\tau_{kv}\right).$$

The updates are

$$\lambda_{dk} \leftarrow \lambda_{dk} + \rho\left[N_d\gamma_{dnk} + \alpha - (N_d + K\alpha)\theta_{dk}\right], \tag{22}$$

$$\tau_{kv} \leftarrow \tau_{kv} + \rho\left[\hat{\gamma}_{kv} + \beta - (\sum_v \hat{\gamma}_{kv} + V\beta)\phi_{kv}\right],$$

where $\hat{\gamma}_{kv} = \frac{D}{M}\sum_{d\in D}\sum_n \mathbb{I}(w_{dn} = v)\gamma_{dnk}$.

We also implement SVRG-based [19] variance reduction for SMD and RSGD, denoting their variance-reduced version as SMD-vr and RSGD-vr. We compare SMD and RSGD with sEM in Fig. 3 for PLSA. The gradient based algorithms converges slower than SEM, which has an exact M-step. Moreover, SMD and RSGD almost make no progress on the large Wiki and PubMed datasets, because of the bad scaling of the gradient. Take RSGD (Eq. 22) as an example, the gradient is proportional with the document length $N_d$. The document length can vary greatly, from less than ten to thousands. Therefore, the parameters for long documents changes faster than short documents due to the larger gradient. If the learning rate is large, the gradients of long documents can be so large that the update is not stable. Therefore, the learning rate is limited by the length of the longest document, and all the other shorter documents will converge slowly. In contrast, sEM updates do not have this problem because all the documents forgets the past sufficient statistics at the same rate. SMD and RSGD can be improved with better tuning of the learning rate, such as line search and adaptive learning rates [9]. However this significantly complicates the implementation, and how to apply variance reduction to these algorithms are unclear.

(a) NYTimes $K = 50$

(b) NYTimes $K = 100$

(c) Wiki $K = 50$

(d) Wiki $K = 100$

(e) PubMed $K = 50$

(f) PubMed $K = 100$

Figure 3: PLSA convergence experiments.