[Reviews · NeurIPS 2018]

Reviewer 1



# After authors response I thank the authors for their response, and for taking into account my remarks about some typos in the mathematical developments and the bounds. # Summary This paper develops a new algorithm based on the EM. It builds on the classical stochastic EM of Cappé and Moulines (2009), with an extra variance reduction term in the iteration formula. This variance reduction technique is inspired from the stochastic gradient descent literature (in particular, algorithms developed in Le Roux et al 2012, Johnson and Zhang 2013, Defazio et al 2014). After setting up the background to present previous results in a unified and clear way, the authors present their algorithm and show two theoretical properties of it: a local convergence rate, and a global convergence property. In the last section, they compare their new algorithm to several state-of-the-art methods, on a Gaussian mixture toy example, and on a probabilistic latent semantic analysis problem. # General comments The article is clearly written, and the main ideas and developments are easy to follow. The problem is nicely presented, and the general framework set up in section 2 helps laying the ground for the two main results and proofs. See specific comments below on some possible typos in the appendices part of the proofs. In practice, the algorithm seems usable, and in particular does not need a sequence of decreasing weights to be calibrated. The Gaussian mixture example gives a useful intuition of how the method works, compared to the standard stochastic EM. I'm not familiar with the literature on latent semantic analysis, and I did not go over the mathematical and methodological details of the main example of section 4.2. The results seem to indicate that the new algorithm performs well, although I have no intuition on what a "significant improvement" is for this problem and criterion. To conclude, this paper presents an algorithm with convergence guaranties that seem theoretically sound, and show how it can be used on a concrete example. I think it has the potential to open new developments path and further improvements for the family of EM-like algorithms. # Specific remarks ## Possible typo in Th. 2 I could not understand the last line of equation in the block between lines 336 and 437. More precisely, I don't understand how the sum on the right can be bounded by M\Delta_e^2. It does not seem at first sight that ||\hat{\theta}_{e, t} - \hat{\theta}_{e, 0}|| is bounded by \Delta = \max_{t} ||\hat{\theta}_{e, t} - \hat{\theta}_{e, t+1}||, which is the only way I could explain the bound. But maybe I'm missing something ? With a naive approach, the best I could get for this bound is to write a triangular inequality, so that: ||\hat{\theta}_{e, t} - \hat{\theta}_{e, 0}|| \leq \sum_{u=0}^{t-1}||\hat{\theta}_{e, u+1} - \hat{\theta}_{e, u}|| \leq t \Delta_e Taking the sum in t, this would give a bound in M(M+1)/2 \Delta_e^2 (instead of M \Delta_e^2). If that's true, then the bound on \rho l.347 should be modified accordingly, as well as l.185 in theorem 2. [\gamma/(M(M+1) L_\eta L_f) instead of \gamma/(2M L_\eta L_f)] I don't think that this modified bound would change the spirit or relevance of th.2. Again, I could be missing a simple argument here, but if that's the case, maybe adding a few words of explanations at this point of the proof could help. ## Possible Math Typos * l.184: specify \gamma-convex *for any x, h* ? * l.184: is p(x, h; \theta) different from p(x, h | \theta) ? The first notation seems to be used everywhere else in the text. * l.418-420: the 128 factor is appearing at the wrong place in the intermediate equations (but this does not affect the final result). If I'm not mistaken, it should be '128 L_f^2' instead of 'L_f^2 / 128' everywhere. So: * \rho_* = … = log(\lambda^2 M / (128 L_f^2))/(\lambda M) [eq. between l.418 and 419, and l.419] * \kappa^2 = 128 L_f^2 / \lambda^2 [l.419, coherent with the main text at l.150] * x = \lambda^2M/(128 L_f^2) [l.420] * l.425 and 429: Should it be the expectation under p(h_i | x_i, \hat{\theta}_{e, 0}), instead of p(h_i, x_i | \hat{\theta}_{e, 0}) ? * l.433 and 434: missing transpose on the differences of \eta ? (3rd line in the first block, 2de line in the second block) * l.434: Replace both Q(\hat{\theta}_{e, t(+1)}) by Q(\hat{\theta}_{e, t(+1)}; \hat{\theta}_{e, 0}) ? * l.435: when invoking the \gamma-strong concavity, maybe it would help to recall that this works because \nabla \hat{Q}_{e, t+1} (\hat{\theta}_{e, t+1}) = 0 ? (If that's indeed key ? I know it would have saved me some time) ## Other Possible Typos * l.165: utilize -> utilizing ? * l.432 assume -> assuming ? * l.23 and other: I think it should be "Cappé", and not "Cappe". ## Reproducibility The algorithms and developments used in section 4 are described at length in the supplementary material, apparently providing all the tools to reproduce the results (but I did not go over these developments closely). However, no ready-to-use package or script seem to be provided to ease this task.

Reviewer 2



The paper introduces a variance reduction strategy for stochastic EM inference. The general approach consist in considering EM as a fix point algorithm and then apply an already proposed a variance reduction technique to solve the fix-point equation. The paper is clear and well written. The context and state of the art are well exposed. The main novelty is to apply the reduced variance step to the EM algorithm, which changes significantly the probabilistic framework. The proposed algorithm achieves a convergence rate that is comparable to the batch EM, which does not always scale for large datasets. Both Theorems 1 and 2 are new, although Thm 1 is obviously the main result. The authors are a bit elliptic about condition (b) of Thm 1. It would be interesting to know for which typical model it holds and for which it does not. It would also be interesting to know if it holds for the models considered in the simulation study.

Reviewer 3



The paper proposes an application of the popular variance reduction construction from the optimization community to stochastic EM (sEM). The authors provide local and global convergence discussions (the latter for strongly concave loglikelihoods); the experiments show some improvement over sEM in latent semantic analysis and LDA. Quality ===== The work is of good quality. The algorithm and theoretical contributions are solid, especially the proof of theorem 2. The experimental section is very well described and should enable easy reproducibility. Clarity ===== The writing quality is very good. The motivation and description of the techniques are easy to follow. Minor: - bug in the citation on line 96. Originality ======== Moderate. The VR construction and local convergence proof are well known; the latter is essentially identical to existing treatments in the optimization literature. The proof of Theorem 2 has some novelty, though the strong convexity assumption makes it intuitively obvious. Significance ========== Overall I think this paper provides a timely application of the VR methodology to stochastic EM. Though not particularly novel, the VR augmentation does exhibit significant improvement in the synthetic experiments and the NYTimes LDA results. There are consistent improvements over standard stochastic EM in the pLSA/LDA experiments, and the gains over the online EM algorithm are impressive. Overall I think the paper provides a useful application of the variance reduction method; while there is no great theoretical novelty, the experimental results should be of interest to practitioners.